# Track Structure of Light Ions: The Link to Radiobiology

**DOI:** 10.3390/ijms24065826

**Published:** 2023-03-18

**Authors:** Valeria Conte, Anna Bianchi, Anna Selva

**Affiliations:** INFN-LNL, Viale dell’Università 2, I-35020 Legnaro, Italy; anna.bianchi@lnl.infn.it (A.B.); anna.selva@lnl.infn.it (A.S.)

**Keywords:** track structure, Monte Carlo simulations, nanodosimetry, radiobiology

## Abstract

It is generally recognized that the biological response to irradiation by light ions is initiated by complex damages at the DNA level. In turn, the occurrence of complex DNA damages is related to spatial and temporal distribution of ionization and excitation events, i.e., the particle track structure. It is the aim of the present study to investigate the correlation between the distribution of ionizations at the nanometric scale and the probability to induce biological damage. By means of Monte Carlo track structure simulations, the mean ionization yield $M_{1}$ and the cumulative probabilities $F_{1}$, $F_{2}$, and $F_{3}$ of at least one, two and three ionizations, respectively, were calculated in spherical volumes of water-equivalent diameters equal to 1, 2, 5 and 10 nm. When plotted as a function of $M_{1}$, the quantities $F_{1}$, $F_{2}$ and $F_{3}$ are distributed along almost unique curves, largely independent of particle type and velocity. However, the shape of the curves depends on the size of the sensitive volume. When the site size is 1 nm, biological cross sections are strongly correlated to combined probabilities of $F_{2}$ and $F_{3}$ calculated in the spherical volume, and the proportionality factor is the saturation value of biological cross sections.

## 1. Introduction

It is generally accepted that the biological response to irradiation with light ions is initiated, to a greater part, by complex lesions at the DNA level [1]. It is also recognized that the complexity of these lesions and their repairability are related to the clustering of energy deposition events at the nanoscopic level and therefore to the particle track structure, i.e., to the spatial and temporal distributions of ionization and excitation events [2,3]. In particular, it can be assumed: (i) that the severity of radiation damage to the DNA increases with an increasing number of particle interactions therein, and (ii) that the ability of cells to recognize and correctly repair the damage decreases with increasing damage complexity. The complete description of particle track structure can be studied by means of Monte Carlo simulations [4], but experimental techniques are not yet able to detect very small events of energy deposition, which are common in volumes of nanometer size. The experimental determination of the stochastic aspects of particle interaction is therefore limited to the ionization component of the energy deposition event. When the number of ionizations is large, it is assumed that the mean ionization yield is proportional to the mean energy deposited and that the stochastic distribution of the number of ionizations produced in the sensitive volume is a good representation of the stochastics of energy deposition. On the other hand, when the number of ionizations is very small, the discrete distribution of the number of ionizations (hereafter called the ionization cluster-size distribution, ICSD) fails in describing the details of the continuous distribution of the energy deposited [5]. The stochastic distribution of energy deposition events derived from ionization measurements (multiplying the number of ionizations by the average energy expended per ion pair formed) is a discrete distribution instead of a continuous one. However, a strong correlation still exists between the moments of the energy and ionization distributions [6]. At the same time, a recent study on the nature of damaging events (measurable ionizations vs. other inelastic channels) induced in simulated DNA targets by swift carbon ion beams in a wide energy range concluded that about 70% of the events leading to the damaging clusters correspond to ionization processes [7]. Based on these findings, and in view of establishing a procedure to characterize the radiation quality of particle tracks based on measurable quantities, the ionization component of particle track structure has been studied both experimentally and by means of Monte Carlo simulations [8,9,10,11,12,13,14,15].

The number *ν* of ionizations created in a nanometer-sized target volume by single ionizing particles penetrating through or passing nearby the target at specified impact parameter *d* is measured, and the corresponding relative frequency $P_{\nu }(Q|d$) represents the probability of cluster size *ν* due to ionizing particles of radiation quality *Q* (defined as the charge state and velocity of a particle) [9]. In this work, only central passage of a spherical target volume V is considered, with primary ions traveling along the V diameter with the impact parameter set to zero. Hence, the dependence on the impact parameter is not made explicit in the notation (being always *d* = 0), while the dependence on the site diameter *D* is explicitly introduced as $P_{\nu }(Q|D$). The chosen irradiation geometry is not meant to mimic the real scenario, where a biological target are obviously irradiated at different impact parameters. The aim is to identify characteristic quantities of the particle track structure that correlate to the radio-induced biological damage better than the particle LET. Based on the probabilities $P_{\nu }(Q|D$), the mean ionization-cluster size caused by single ionizing particles in the target volume is given by the moment $M_{1}(Q|D)$, which is defined by Equation (1) for ξ= 1.
(1)$$M_{\xi }(Q|D)=\overset{\infty }{\underset{\nu =0}{\sum }}\nu^{\xi }P_{\nu }(Q|D)$$

Similarly, the complementary cumulative probability of forming ionization-cluster sizes ν≥k is given by the sum defined by Equation (2).
(2)$$F_{k}(Q|D)=\overset{\infty }{\underset{\nu =k}{\sum }}P_{\nu }(Q|D)=1-\overset{k-1}{\underset{\nu =0}{\sum }}P_{\nu }(Q|D)$$

Here, $F_{1}(Q|D$) represents the probability that an ionizing particle creates at least one ionization within the target volume, and $F_{2}(Q|D$) is the probability that an ionizing particle creates at least two ionizations in it. The cumulative probabilities $F_{k}(Q|D)$ describe the clustering of ionizations within short distances and are therefore related to the induced radiobiological DNA damage. Based on a large amount of data, it has been observed that when plotted as a function of $M_{1}$, the quantities $F_{1}$, $F_{2}$ and $F_{3}$ behave as almost unique curves, independent of radiation quality, and correlate with the biological inactivation cross sections, if measured in target volumes between 1 and 1.5 nm in size [16,17]. While it was recognized that the contribution of the *δ*-electrons depends on site size, it was inferred that this variable contribution has only a minor influence on the functional dependency of $F_{k}$ on $M_{1}$, at least for k=1, 2, 3 . The aim of this work was to study in more detail the shape of the curves $F_{k}(M_{1}$) for different sizes of the target volume and their correlation to the inactivation of cross sections for V79 and HSG cells available in the literature.

This work reports on Monte Carlo track-structure simulations [9,18] performed for protons, helium and carbon ions of different energies interacting with gaseous spherical target volumes equivalent to water spheres of diameters 1, 2, 5 and 10 nm [19]. The traveling direction of primary particles was set along the sphere diameter (impact parameter set to zero).

## 2. Results

### 2.1. Track Structure Characteristics

Figure 1 shows the cumulative distributions $F_{1}$, $F_{2}$ and $F_{3}$ simulated in target volumes of diameters 1, 2, 5, and 10 nm in water, as a function of the mean ionization yield $M_{1}$. Tables with the simulated data are given in Appendix A. It can be observed that, for a specified site size, $F_{k}$ depends almost exclusively on $M_{1}$, independent of particle type (protons, helium and carbon ions) and velocity. However, the shape of the curve depends on site size: at small $M_{1}$ values, the difference between $F_{1}$, $F_{2}$, and $F_{3}$ is maximum at diameter *D* = 1 nm and decreases with increasing site sizes. The cumulative distributions show a saturation effect at increasing $M_{1}$, which follows from their definition. The variability between the smallest value of $F_{2}$, corresponding to 200 MeV protons, and the largest one, corresponding to 1 AMeV carbon ions, is of about 3.5, 3.0, 2.5 and 2 orders of magnitude at 1, 2, 5 and 10 nm, respectively. The variation is even more pronounced for $F_{3}$, while it is less evident for $F_{1}$.

In Figure 1, symbols correspond to ion types for which the Monte Carlo simulations were performed. The lines are the best fit of the simulated data, performed with the following equations, which offer a parametrization of $F_{k}\left(M_{1}|D\right)$ and will be discussed later:(3)$$F_{1}\left(M_{1}|D\right)=1-e^{-C_{1}\left(D\right)M_{1}}$$
(4)$$F_{2}\left(M_{1}|D\right)=1-\left[1+C_{2}\left(D\right)M_{1}\right]e^{-C_{1}\left(D\right)M_{1}}$$
(5)$$F_{3}\left(M_{1}|D\right)=1-\left[1+C_{3}\left(D\right)M_{1}+\frac{C_{2}^{2}\left(D\right)}{2}M_{1}^{2}\right]{\;e}^{-C_{1}\left(D\right)M_{1}}$$

The values of the fitting parameters $C_{1}\left(D\right)$, $C_{2}\left(D\right)$ and $C_{3}\left(D\right)$, which depend on the diameter D of the simulated site but are independent of particle type, are reported in Table 1.

To make the functional dependency of $F_{k}$ on $M_{1}$ at the different simulated site sizes more clear, Figure 2 shows $F_{1}(M_{1}|D)$ in panel (a), $F_{2}(M_{1}|D)$ in panel (b) and $F_{3}(M_{1}|D)$ in panel (c). The black lines represent the values that would derive from a pure Poisson distribution of the cluster size *ν*, with mean value $M_{1}$. It can be observed that $F_{1}(M_{1}|D)$ always lies below the corresponding “Poisson-type” curve, while $F_{2}\left(M_{1}|D\right)$ and $F_{3}(M_{1}|D)$ always lie above. The deviations increase with increasing site size, reflecting the growing contribution by *δ*-electrons to the total ionization inside the target volume. When the site size is small, the contribution by *δ*-electrons is also small, and the stochastics of the number of ionizations is dominated by the component due to ionizations of the primary particle, which is Poisson distributed.

It is clear from Figure 2 that the cumulative probabilities $F_{k}(M_{1}|D)$ have a functional dependency on $M_{1}$ that depends on the size of the simulated volume, in particular for small sizes with D≤5 nm.

The complete $P_{\nu }(Q|D)$ distributions were simulated, including the probability for cluster size ν=0. This choice is different from that by other authors [20,21,22], who instead considered the conditional distributions, i.e., without taking into account the passage of a particle that does not produce any ionization in the target volume. The reason to keep count of the zeroes is that the complete distributions $P_{\nu }(Q|D)$ have the additive property, i.e., in a mixed radiation field, the resulting $P_{\nu }\left(Q|D\right)$ distribution is the sum of all the $P_{\nu }^{\left(j\right)}\left(Q|D\right)$ corresponding to all the (*j*) components, each weighted with the relative frequency $\phi^{\left(j\right)}$ of radiation quality (*j*). The same additive property holds for all the derived quantities, in particular for the moments $M_{\xi }\left(j\right)$ and the cumulative quantities $F_{k}\left(j\right)$.

### 2.2. The Link to Radiobiology

It has already been noted [16], and it is confirmed in this study, that the cumulative distribution functions $F_{k}(Q|D)$ behave, when plotted as a function of mean cluster size $M_{1}$, in a similar way as biological inactivation cross sections [23] plotted as a function of LET: they first increase with increasing $M_{1}$ or LET until they show a saturation effect at large $M_{1}$ or LET values. In particular, a strong correlation has been observed between the cumulative probability $F_{2}(Q|D)$ and the inactivation cross section at 5% survival, $\sigma_{5\%}$ [17]. To study this correlation in more detail, biological data for V79 asynchronous cells irradiated by protons, helium and carbon ions, available in the PIDE radiobiological database [24], are plotted in Figure 3b versus the ratio $\left(1\;\mathrm{nm}\right)/\lambda_{ion}$, where $\lambda_{ion}$ is the mean free ionization path length of the primary particles in propane [9]; the ratio represents the mean number of ionizations produced by the primary particle in 1 nm path length. In Figure 3a, the probabilities $F_{2}(Q|D)$ calculated in simulated sites of diameter *D* equal to 1 and 2 nm are also plotted versus the ratio $\left(1\;\mathrm{nm}\right)/\lambda_{ion}$.

It can be easily observed that the best correspondence is achieved for *D* = 1 nm. This result is clearer in Figure 4, where the inactivation cross sections $\sigma_{5\%}$ for carbon ion irradiation are plotted as a function of $F_{2}(Q|D)$ for different site sizes: 0.5, 1 and 2 nm. The best correlation between $\sigma_{5\%}$ and $F_{2}(Q|D)$ is found when $F_{2}(Q|D)$ is measured in a simulated volume of diameter *D* = 1 nm. If the site size is smaller, for instance *D* = 0.5 nm, the cross sections $\sigma_{5\%}$ bend, and their growth slows for $F_{2}(Q|D)>0.5$ when saturation begins to take effect and the correlation is lost. If the target volume is larger than 1 nm, for instance *D* = 2 nm, $F_{2}(Q|D)$ reaches the saturation value of $F_{2}\left(Q|D\right)=1$ too early, for radiation qualities at which biological cross sections are still growing. For *D* = 1 nm, $\sigma_{5\%}$ has good approximation proportional to $F_{2}\left(Q|D\right)$ for all radiation qualities, the proportionality factor being *k* = 50 μm^2^, which approximately corresponds to the saturation value of the biological cross sections.

The correlation between biological cross sections and nanodosimetric quantities was therefore investigated in detail for a simulated target volume 1 nm in diameter. The correlations of $\sigma_{5\%}$ and $\sigma_{\alpha }$ with specific combinations of the cumulative probabilities $F_{2}$ and $F_{3}$ are represented in Figure 5a,b, respectively. The relative residuals, plotted in the bottom panels, are Gaussian distributed with a standard deviation of 0.18 for $\sigma_{5\%}$ and 0.35 for $\sigma_{\alpha }$.

The analysis was extended to another cell line, specifically to asynchronous HSG cells irradiated by helium and carbon ions, data which are also available in the PIDE database. The results are illustrated in Figure 6, which shows the biological cross sections $\sigma_{5\%}$ (panel a) and $\sigma_{\alpha }$ (panel b) plotted as a function of the linear combinations $\left(0.8F_{2}+0.2F_{3}\right)$ and $\left(0.2F_{2}+0.8F_{3}\right)$, respectively. Similarly to the V79 cells, a good correlation was found but with a proportionality factor *k* = 0.75 μm^2^, reflecting the larger sensitive volume for HSG compared to that of V79 cells, also indicated in other works [25].

## 3. Discussion

### The Probabilistic Description of Ionization–Cluster Size Formation

The behavior of the cumulative distribution functions $F_{k}(Q|D)$ at different site sizes can be interpreted on the basis of a probabilistic theory of ionization–cluster size formation [9]. For ionizing particles of radiation quality *Q* crossing a nanometer-sized spherical target volume *V* along its diameter *D*, it is assumed that the ionization cluster-size caused within *V* is exclusively determined by the average number $\bar{\kappa (Q|D)}$ of ionizing interactions of a primary particle along *D* and by the behavior of *δ*-electrons within the target volume.

Based on these basic assumptions, the probability $P_{\nu }(Q|D)$ of ionization–cluster size *ν* is given by a compound Poisson process [9] described by Equation (6).
(6)$$P_{\nu }\left(Q|D\right)=\overset{\infty }{\underset{\kappa =0}{\sum }}\frac{{\bar{\kappa \left(Q|D\right)}}^{\kappa }{\;e}^{-\bar{\kappa \left(Q|D\right)}}}{\kappa !}\times p_{\nu }^{\left(\kappa \right)}\left(Q|D\right)$$

Here, $p_{\nu }^{\left(\kappa \right)}\left(Q|D\right)$ is the probability that in the event of exactly *κ* primary ionizations due to primary particles of radiation quality *Q*, a cluster size *ν* is formed within the target volume (the ionizations due to *δ*-electrons are included). The average number $\bar{\kappa (Q|D)}$ of ionizing interactions of a primary particle along *D* is given as the quotient $\bar{\kappa \left(Q|D\right)}=\left(D\rho \right)/{\left(\lambda \rho \right)}_{\mathrm{ion}}$, where ${\left(\lambda \rho \right)}_{\mathrm{ion}}$ is the mean free ionization path length of the primary particles in matter.

In view of the fact that in a short track segment, an ionization process due to a primary particle is independent of the number of previously formed ions, the $p_{\nu }^{\left(\kappa \right)}\left(Q|D\right)$ distribution is given by the *κ*-fold convolution of the probability distribution $p_{\nu }^{\left(\kappa \right)}\left(Q|D\right)$, *ν* = 1, 2, 3, …, in the case of a single primary ionization (*κ* = 1), which is referred to in the following single-ionization distribution, i.e.,
(7)$$p_{\nu }^{\left(\kappa \right)}\left(Q|D\right)=p_{\nu }^{\left(1\right)}\left(Q|D\right)*p_{\nu }^{\left(1\right)}\left(Q|D\right)*\ldots *p_{\nu }^{\left(1\right)}(Q|D)$$
where the convolution operation, indicated by the asterisk, is performed *κ* times (*κ*-fold convolution) and is defined for two discrete functions $f_{\nu }$ and $g_{\nu }$, *ν* = 0, 1, 2, …, as ${\left(f*g\right)}_{\nu }=\sum_{µ=0}^{\nu }f_{\nu -µ}g_{µ}$.

The single-ionization distributions $p_{\nu }^{\left(1\right)}(Q|D)$, *ν* = 0, 1, 2, … represent the cluster-size formation due to a single primary ionization event; therefore, they are independent of $\bar{\kappa \left(Q|D\right)}$, but at least in principle, they depend on the spectral distribution of secondary electrons set in motion by impact ionization and, thus, on the particle’s velocity. In contrast,$\;\bar{\kappa (Q|D)}$ is determined by the mean free ionization path length ${\left(\lambda \rho \right)}_{\mathrm{ion}}$ of the primary particles in matter and, thus, on the particles’ charge state and velocity.

To relate the ionization–cluster size probabilities defined by Equation (6) to the members of the single-ionization distribution, the formalism of folding discrete distributions can be applied:(8)$$p_{\nu }^{\left(\kappa \right)}(Q|D)=\left\{\begin{array}{c} \delta_{0\nu }\;\;\mathrm{for}\;\kappa =0\\ \overset{\nu }{\underset{j=0}{\sum }}p_{\nu -j}^{\left(\kappa -1\right)}(Q|D)\times p_{j}^{\left(1\right)}(Q|D)\;\;\mathrm{for}\;\kappa \geq 1 \end{array}\right.$$

Here, Equation (8) represents the folding of the single-ionization distribution $p_{\nu }^{\left(1\right)}\left(Q|D\right)$ with the distribution $p_{\nu }^{\left(\kappa -1\right)}\left(Q|D\right)$ in the case of exactly *κ*-1 primary ionizations. For *κ* = 0, the expression $\delta_{0\nu }$ reflects the fact that, if no primary ionizations take place, no *δ*-electrons are produced in the target volume, and the only possible cluster size is *ν* = 0. This assumption neglects the contribution to the total ionization by *δ*-electrons that are produced outside of the target volume *V* and enter it. For *κ* ≥ 1, by successive application of the convolution operation, all members of the $p_{\nu }^{\left(\kappa \right)}(Q|D)$-distribution can be written as a sum of products, which exclusively consists of members of the $p_{\nu }^{\left(1\right)}(Q|D)$-distribution in the case of a single primary ionization. For particles directly crossing the target volume *V*:(9)$$p_{\nu }^{\left(k\right)}\left(Q|D\right)=0\;\mathrm{for}\;k>\nu$$
because in volume *V*, there are at least k primary ionizations. As a consequence, in case of a particle traversing the target volume, the superior limit of the summation in Equation (8) can be substituted by ν:(10)$$P_{\nu }\left(Q|D\right)=\overset{\nu }{\underset{\kappa =0}{\sum }}\frac{{\bar{\kappa \left(Q|D\right)}}^{\kappa }{\;e}^{-\bar{\kappa \left(Q|D\right)}}}{\kappa !}\times p_{\nu }^{\left(\kappa \right)}(Q|D)$$

The first values of $P_{\nu }\left(Q|D\right)$, for ν=0, 1, 2, can be easily obtained:(11)$$P_{0}\left(Q|D\right)=e^{-\bar{\kappa (Q|D)}}$$
(12)$$P_{1}\left(Q|D\right)=\bar{\kappa \left(Q|D\right)}e^{-\bar{\kappa (Q|D)}}\;p_{1}^{\left(1\right)}\left(Q|D\right)$$
(13)$$P_{2}\left(Q|D\right)=\left\{\bar{\kappa \left(Q|D\right)}\;p_{2}^{\left(1\right)}\left(Q|D\right)+\frac{{\bar{\kappa \left(Q|D\right)}}^{2}}{2}{\left[p_{1}^{\left(1\right)}\left(Q|D\right)\right]}^{2}\right\}{\;e}^{-\bar{\kappa \left(Q|D\right)}}$$

The cumulative distributions $F_{1}(Q|D)$ and $F_{2}(Q|D)$ and $F_{3}(Q|D)$ can be simply derived and written in terms of the members $p_{1}^{\left(1\right)}(Q|D)$ and $p_{2}^{\left(1\right)}\left(Q|D\right)$ of the single-ionization distribution:(14)$$F_{1}\left(Q|D\right)=1-e^{-\bar{\kappa \left(Q|D\right)}}$$
(15)$$F_{2}\left(Q|D\right)=1-\left[1+\bar{\kappa (Q|D)}\;p_{1}^{\left(1\right)}(Q|D)\right]e^{-\bar{\kappa (Q|D)}}$$
(16)$$F_{3}\left(Q|D\right)=1-\left\{1+\bar{\kappa \left(Q|D\right)}\left[\;p_{1}^{\left(1\right)}\left(Q|D\right)+p_{2}^{\left(1\right)}\left(Q|D\right)\right]+\frac{{\bar{\kappa \left(Q|D\right)}}^{2}}{2}{\left[p_{1}^{\left(1\right)}\left(Q|D\right)\right]}^{2}\right\}{\;e}^{-\bar{\kappa (Q|D)}}$$

$M_{1}(Q|D)$ can also be written in terms of $\bar{\kappa \left(Q|D\right)}$ and the mean value $m_{1}(Q|D)$ of the $p_{\nu }^{\left(1\right)}\left(Q|D\right)$ distribution, as derived in detail in [9]:(17)$$M_{1}\left(Q|D\right)=\bar{\kappa \left(Q|D\right)}\;m_{1}(Q|D)$$

It can be observed that, under the hypothesis $p_{\nu }^{\left(0\right)}=\delta_{0\nu }$, $F_{1}\left(Q|D\right)$ depends only on the quotient $D/{\left(\lambda \rho \right)}_{\mathrm{ion}}$, while $F_{2}\left(Q|D\right)$ and $F_{3}\left(Q|D\right)$ depend on the single ionization distribution values $\;p_{1}^{\left(1\right)}$ and $\;p_{2}^{\left(1\right)}$ and therefore on the spectral distribution of secondary *δ*-electrons. However, Figure 1 shows only negligible dependency of $F_{1}\left(Q|D\right)$, $F_{2}\left(Q|D\right)$ and $F_{3}\left(Q|D\right)$ on particle type and thus also on the primary particle velocity if $M_{1}\left(Q|D\right)$ is the same.

Comparing Equations (14)–(16) with the fitting Equations (3)–(5), the following relations exist for the fitting parameters $C_{k}\left(D\right)$:(18)$$C_{1}\left(D\right)=\frac{\;\bar{\kappa \left(Q|D\right)}}{M_{1}\left(Q|D\right)}=\frac{1}{m_{1}\left(Q|D\right)}$$
(19)$$C_{2}\left(D\right)=C_{1}\left(D\right)\cdot p_{1}^{\left(1\right)}(Q|D)$$
(20)$$C_{3}\left(D\right)=C_{1}\left(D\right)\left[\;p_{1}^{\left(1\right)}\left(Q|D\right)+p_{2}^{\left(1\right)}\left(Q|D\right)\right]$$

As defined in Equation (18), the reciprocal of $C_{1}\left(D\right)$ represents the mean ionization yield per single primary ionization; thus, it is a measure of the additional contribution by the *δ*-electrons to the primary ionization. According to Table 1, the contribution of *δ*-electrons to the total average ionization yield amounts to about 10%, 17%, 27% and 33% of the total for site sizes of 1, 2, 5 and 10 nm, respectively. Figure 1 shows, in particular, that all $F_{1}\left(Q|D\right)$ values for protons, helium and carbon ions lie on a unique curve. The parameter $C_{1}\left(D\right)$ is independent of particle type and velocity, which can be interpreted in the sense that, on average, each *δ*-electron contributes to the ionization cluster with an additional mean number of ionizations that is largely independent of particle type and velocity. To confirm this finding, Figure 7 shows the cluster size distributions in 1 nm site size, due to the total contribution by primary particles and *δ*-electrons, and that due to ionizations of the primary particle only, for different radiation qualities. The results for the mean cluster sizes and their ratios are given in Table 2. The values of the quotient $\frac{\;\bar{\kappa \left(Q|D\right)}}{M_{1}\left(Q|D\right)}$ found by direct simulation of ICSD for several radiation qualities are almost invariant with particle velocity and are in very good agreement with the value of the parameter $C_{1}\left(D\right)$, given in Table 1 for 1 nm site size. This implies that $m_{1}\left(Q|D\right)$ also depends negligibly on radiation quality.

Equation (19) implies that the probability that the single primary ionization results in a cluster of size *ν* = 1, $\;p_{1}^{\left(1\right)}\left(Q|D\right)$, is also largely independent of particle type and velocity, but depends on site diameter *D*. Recursively, the same conclusion can be drawn for the probability $p_{2}^{\left(1\right)}\left(Q|D\right)$.

In the works by Conte et al. [16,17], a good correlation was found between radiobiological cross sections at 5% survival and $F_{2}$ measured in a volume of 1 nm diameter, and between the cross sections at low doses, $\sigma_{\alpha }$, and $F_{3}$, measured in a volume of diameter 1.5 nm. The proportionality factors were slightly different in the two cases. These results were based on the assumption of a unique dependence of $F_{2}$ and $F_{3}$ on the mean cluster size $M_{1}$, also independent of target size. Consistently, only the values $M_{1}$ were simulated for radiation qualities at which biological data were available, and then, values of $F_{2}$ and $F_{3}$ were assigned, based on experimental results obtained at larger target volumes, neglecting the dependence of the functions $F_{k}\left(M_{1}\right)$ on target size. In this work, more accurate simulations of ionization cluster size distributions and derived cumulative distributions were performed at the different radiation qualities investigated. It was found out that the biological cross sections, $\sigma_{5\%}$ and $\sigma_{\alpha }$ at least for V79 and HSG cells, depend on linear combinations of $F_{2}$ and $F_{3}$ calculated in a simulated spherical volume of 1 nm in diameter:(21)$$\sigma_{5\%}\left(Q\right)=k_{cell}\left[0.8F_{2}\left(Q\right)+0.2F_{3}\left(Q\right)\right]$$
(22)$$\sigma_{\alpha }\left(Q\right)=k_{cell}\left[0.2F_{2}\left(Q\right)+0.8F_{3}\left(Q\right)\right]$$

Note that the proportionality factor $k_{cell}$ is the same for both cross sections and corresponds to their unique saturation value. It is a parameter that depends on the specific cell line and corresponds approximately to the nucleus size; it was found to be 50 μm^2^ for V79 cells and 75 μm^2^ for HSG cells.

In terms of $P_{0}\left(Q\right)$, $P_{1}\left(Q\right)$ and $P_{2}\left(Q\right)$, Equations (20) and (21) can be rewritten as:(23)$$\sigma_{5\%}\left(Q\right)=k_{cell}\left[1-P_{0}\left(Q\right)-P_{1}\left(Q\right)-0.2P_{2}\left(Q\right)\right]$$
(24)$$\sigma_{\alpha }\left(Q\right)=k_{cell}\left[1-P_{0}\left(Q\right)-P_{1}\left(Q\right)-0.8P_{2}\left(Q\right)\right]$$

Equations (23) and (24) express that the biological cross sections are strongly correlated with the probability of cluster sizes *ν* = 0, 1 and 2.

## 4. Materials and Methods

The Monte Carlo simulations of the cluster size distributions were performed with the so-called MC-Startrack model, developed by Grosswendt in 2002 to simulate the experimental response of the Startrack counter [9], and later upgraded in 2014 [14]. It is a homemade code, not freely distributed, capable of simulating the stochastic ionization yield in propane gas volumes, and it can also include the efficiency map of the Startrack counter. It has been validated against a great number of experimental data [17,18]. In this work, the simulations were performed in gaseous spherical volumes filled with propane gas at 300 Pa, for diameters of 0.15, 0.3, 0.75 and 1.5 mm, equivalent to diameters of 1, 2 5 and 10 nm in water [8]. The model assumes that in thin layers of gaseous propane elastic scattering, charge-changing effects and impact excitation processes of the primary particles can be neglected. According to this assumption, the cluster-size distributions caused by ionizing particles penetrating through a specified target volume are determined exclusively by the path lengths of the light ions between successive ionizing interactions, by the spectral and angular distributions of *δ*-electrons set in motion at the interaction points, and by the properties of electron degradation in the target medium. In the present work, the total and the differential ionization cross section for light ions was calculated by applying the model of Rudd et al. [18,26]. For details of the Monte Carlo model and its experimental validation, see References [9,10,18].

The target volume was immersed in a larger interaction volume at the same gas density, the size of which was determined to ensure that *δ*-electrons generated along the primary particle track but outside the sensitive volume can interact with it. A total number of histories was simulated varying between 10^5^ for large mean ionization yields $M_{1}$ and 10^8^ for small values of $M_{1}$.

Calculations were performed on a Windows desktop PC, CPU type: Intel(R) Core(TM) i9-9900K CPU @ 3.60GHz, RAM 64 GB. The biological inactivation cross sections at survival level = *l* were calculated with the following equation, as described in [23]:(25)$$e^{-\sigma_{l}\Phi_{l}}{=e}^{-s_{l}D_{l}}$$
where Φ is the particle fluence, *sl* is the slope of the cell-survival curve in semi-logarithmic scale at dose *D* = *Dl*. In the framework of the linear quadratic model for cell survival, *sl* is given by:(26)$$s_{l}=\alpha +2\beta D_{l}$$

Inactivation cross sections *σ**l* were calculated at initial survival ($s_{l}=\alpha )$ and from the final slope at *l* = 0.05, i.e., at 5% survival:(27)$$\sigma_{l}\left(Q\right)=0.1602\cdot \mathrm{LET}\sqrt{\alpha^{2}\left(Q\right)-4\beta \left(Q\right)\mathrm{Ln}\left(l\right)}$$
where $\sigma_{l}\left(Q\right)$ is given in units of μm^2^ if LET is keV/μm, $\alpha \left(Q\right)$ in Gy^−1^ and $\beta \left(Q\right)$ in Gy^−2^.

To study the correlation of radiobiological data to nanodosimetric quantities, Monte Carlo simulations were performed to calculate, for any specific particle type and energy (as reported in the PIDE database), the ionization cluster size distributions in simulated site sizes of 1, 2, 5 and 10 nm. The first moment $M_{1}(Q|D)$ was calculated afterward.

In this work, data for asynchronous V79 cells irradiated by monoenergetic protons, helium and carbon ions and for asynchronous HSG cells irradiated by monoenergetic helium and carbon ions were taken from the GSI-PIDE library [24].

## 5. Conclusions

Monte Carlo track-structure simulations were performed to simulate the ionization component of particle track structure for protons, helium and carbon ions at different energies, corresponding to different radiation qualities Q. Calculations were performed for spherical volumes filled with low-density propane gas to simulate water spheres of diameters D = 1, 2, 5 and 10 nm, and for particles crossing the target volume with impact parameter set to zero. From the full probabilities of the number of ionizations *ν*, the mean ionization yield $M_{1}(Q|D)$ and the complementary cumulative probability $F_{1}(Q|D)$, $F_{2}(Q|D)$ and $F_{3}(Q|D)$, representing the probabilities of at least 1, at least 2 and at least 3 ionizations, were derived. It was highlighted that the functional dependency of $F_{k}(Q|D)$ on $M_{1}$ clearly depends on the size of the target volume. Parametrizations of $F_{1}(Q|D)$, $F_{2}(Q|D)$ and $F_{3}(Q|D)$ on $M_{1}$ were found that depend only on *D* and not on particle type and velocity. A strong correlation was found between inactivation cross sections for V79 and HSG cells and linear combinations of $F_{2}(Q|D)$ and $F_{3}(Q|D)$ measured in a target volume of diameter D = 1 nm. The unique proportionality factor depends only on cell line and corresponds to the saturation value of biological cross sections.

The nanodosimetric quantities measured for the particles that pass through the target volume at its center (impact parameter set to zero) clearly do not offer a complete description of the radiation interaction at the cellular and sub-cellular levels; similarly, not even the LET or the microdosimetric linear energy distributions do. The purpose of nanodosimetry, and of this work in particular, is to identify measurable physical quantities that characterize the radiation quality in relation to the induced biological damage. The complementary cumulative probabilities $F_{1}(Q|D)$, $F_{2}(Q|D)$ and $F_{3}(Q|D)$, measured in a volume of 1 nm in size for particles’ central passage, seem to be good candidates: they are correlated to inactivation biological cross sections better than LET, at least for V79 and HSG cells irradiated by broad beams of protons, He and C ions.

## Figures and Tables

**Figure 1 ijms-24-05826-f001:**
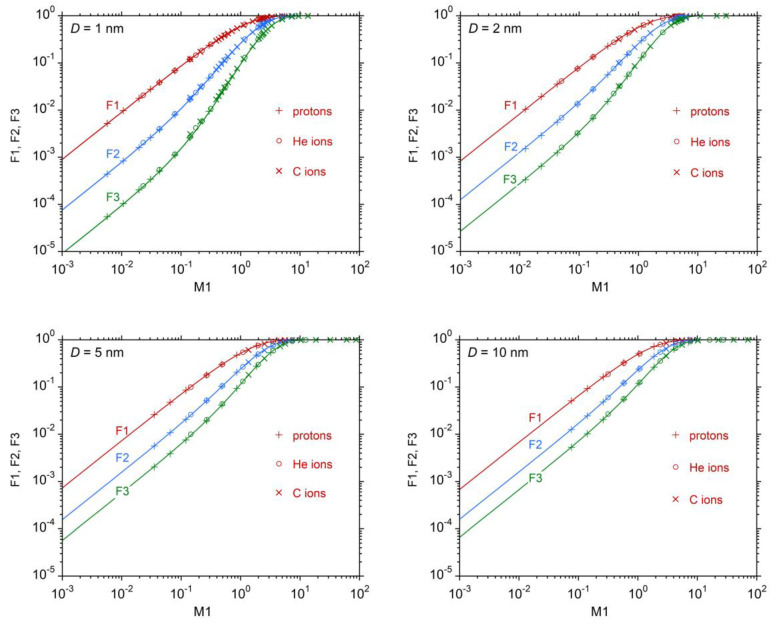
The cumulative probabilities $F_{k}$, for *k* = 1, 2, 3, plotted as a function of the mean cluster size $M_{1}$, for different site sizes. Simulations performed for protons, helium and carbon ions (see legend) at different kinetic energies. Lines are the best fit according to Equations (3)–(5).

**Figure 2 ijms-24-05826-f002:**
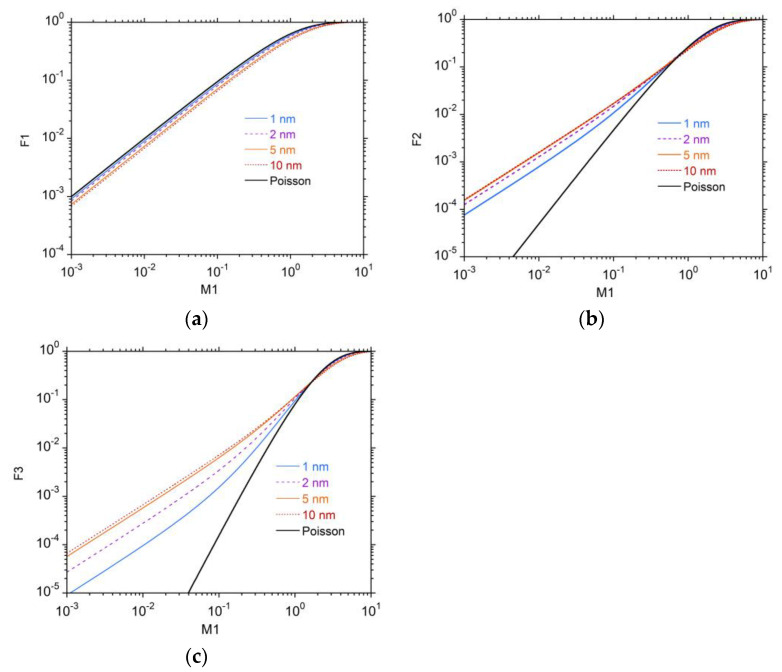
The cumulative probabilities $F_{k}(M_{1}|D)$ calculated at different site sizes and compared with the corresponding values deriving from a pure Poisson distribution for the probability of cluster size *ν*. (**a**) The functional dependency of $F_{1}(M_{1}|D)$. (**b**) The functional dependency of $F_{2}(M_{1}|D)$. (**c**) The functional dependency of $F_{3}(M_{1}|D)$.

**Figure 3 ijms-24-05826-f003:**
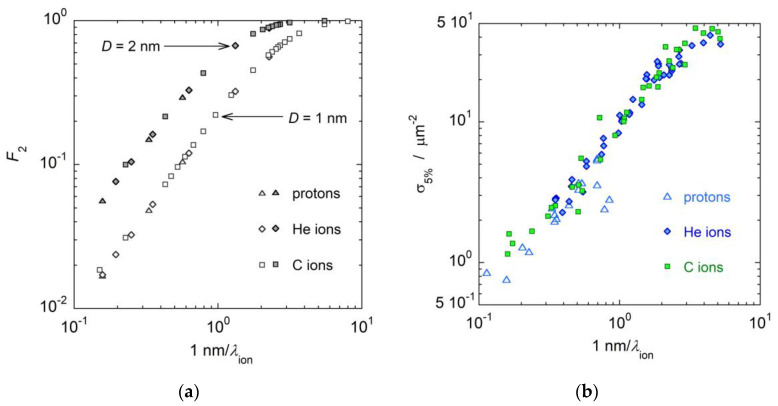
(**a**): The cumulative probabilities $F_{2}\left(Q\right)$, calculated in target volumes with *D* = 1 nm and *D* = 2 nm, are plotted versus the ratio $\left(1\;\mathrm{nm}\right)/\lambda_{ion}$. (**b**) The biological inactivation cross sections at 5% survival level for V79 cells irradiated by protons, helium and carbon ions [17], plotted versus the ratio $\left(1\;\mathrm{nm}\right)/\lambda_{ion}$. See text for more details.

**Figure 4 ijms-24-05826-f004:**
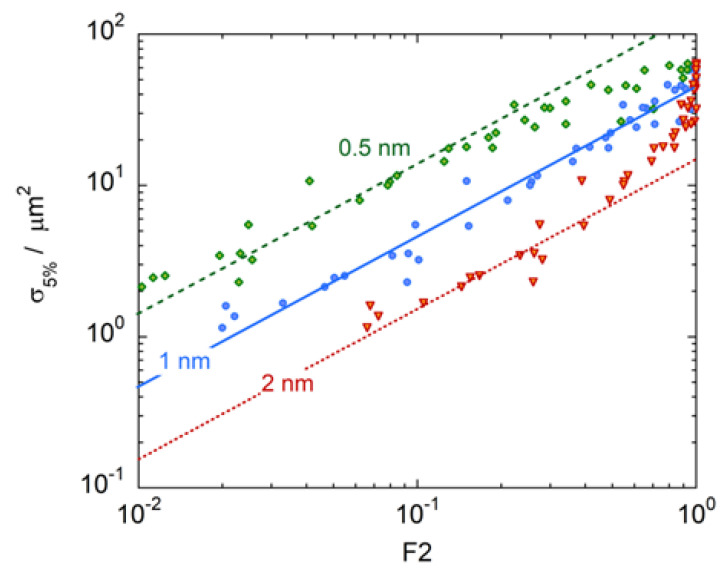
Biological inactivation cross sections at 5% survival, $\sigma_{5\%}$, for asynchronous V79 cells irradiated by carbon ions as a function of the cumulative probabilities $F_{2}\left(Q\right)$ calculated in simulated target volumes of diameters 0.5 (green), 1 (blue) or 2 nm (red). Symbols represent biological cross sections derived from data available in literature [24]. Lines are the linear through zero fits of biological data.

**Figure 5 ijms-24-05826-f005:**
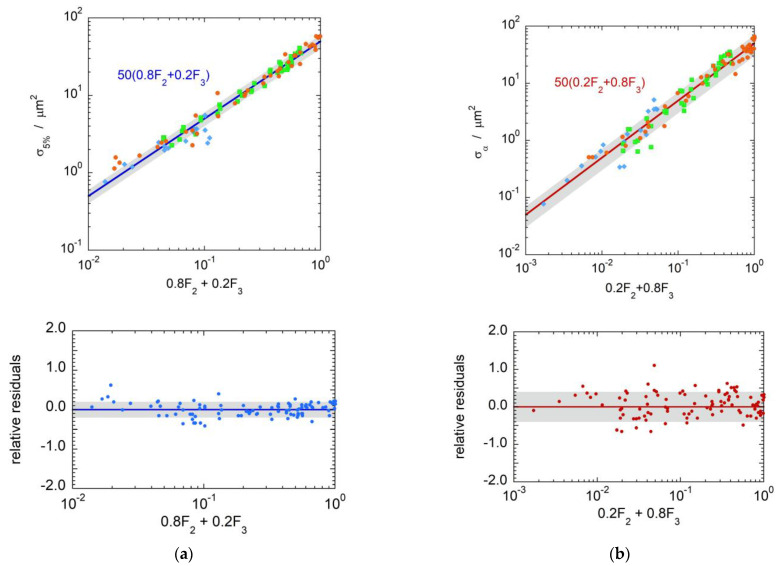
(**a**) Biological inactivation cross sections at 5% survival, $\sigma_{5\%}$, for asynchronous V 79 cells irradiated by protons (diamonds), helium (squares) and carbon ions (circles) as a function of the cumulative probabilities $\left[0.8F_{2}\left(Q\right)+0.2F_{3}\left(Q\right)\right]$ in target volumes of diameter 1 nm. The bottom panel shows the relative residuals with respect to the function $50\left(0.8F_{2}+0.2F_{3}\right)$. (**b**) Biological inactivation cross sections at high survival, $\sigma_{\alpha }$, for asynchronous V79 cells irradiated by protons, helium and carbon ions as a function of the cumulative probabilities $\left[0.2F_{2}\left(Q\right)+0.8F_{3}\left(Q\right)\right]$ in a target volume of diameter 1 nm. The bottom panels (**a**,**b**) show the relative residuals of $\sigma_{5\%}$ and $\sigma_{\alpha }$ with respect to the functions $50\left[0.8F_{2}+0.2F_{3}\right]$ and $50\left[0.2F_{2}+0.8F_{3}\right]$, respectively.

**Figure 6 ijms-24-05826-f006:**
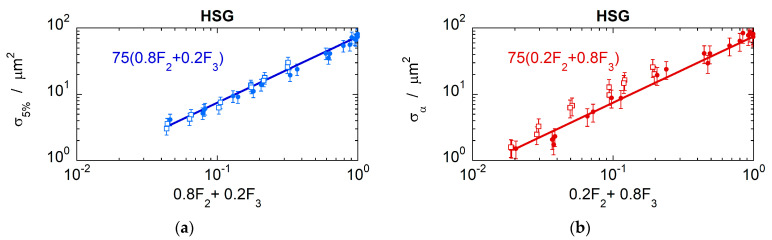
(**a**) Biological inactivation cross sections $\sigma_{5\%}$ for asynchronous HSG cells irradiated by helium (empty squares) and carbon ions (filled circles) as a function of the cumulative probabilities $0.8F_{2}\left(Q\right)+0.2F_{3}\left(Q\right)$ in a target volume of diameter *D* = 1 nm. (**b**) Biological inactivation cross sections at high survival, $\sigma_{\alpha }$, for asynchronous HSG cells irradiated by helium (empty squares) and carbon ions (filled circles) as a function of the cumulative probabilities $0.2F_{2}\left(Q\right)+0.8F_{3}\left(Q\right)$ in a simulated target volume of diameter *D* = 1 nm.

**Figure 7 ijms-24-05826-f007:**
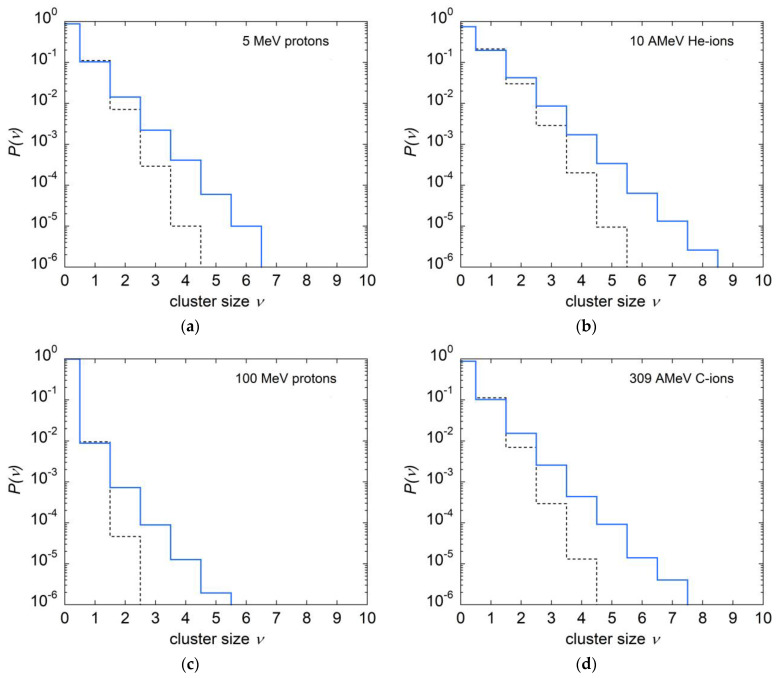
Total ionization cluster size distributions (blue lines) and distributions of the number of ionizations due to direct interaction of the primary particles (dashed lines) for (**a**): 5 MeV protons, (**b**): 10 AMeV He-ions, (**c**): 100 MeV protons and (**d**): 309 AMeV carbon ions.

**Table 1 ijms-24-05826-t001:** The parameters $C_{k}\left(D\right)$ of the fitting functions given in Equations (3)–(5) for the different site sizes.

Diameter *D* *	$C_{1}\left(D\right)$	$C_{2}\left(D\right)$	$C_{3}\left(D\right)$
1 nm	0.907	0.831	0.898
2 nm	0.831	0.705	0.804
5 nm	0.735	0.579	0.678
10 nm	0.670	0.520	0.612

* The target volume diameter is given in terms of the equivalent water sphere.

**Table 2 ijms-24-05826-t002:** Total mean ionization yield $M_{1}$, mean number of ionizations $\bar{\kappa \left(Q\right)}$ due to primary interactions only, and ratio $\bar{\kappa \left(Q\right)}/M_{1}$ for several radiation qualities in a target volume of 1 nm.

Q	$M_{1}$	$\bar{\kappa \left(Q\right)}$	$\bar{\kappa \left(Q\right)}/M_{1}$
1H 5 AMeV	0.14090	0.12670	0.899
4He 10 AMeV	0.31640	0.28300	0.894
He 25.7 AMeV	0.14030	0.12640	0.901
1H 50 AMeV	0.01963	0.01769	0.901
1H 100 AMeV	0.01065	0.00961	0.902
12C 309 AMeV	0.14291	0.12692	0.888

## Data Availability

Nanodosimetric characteristics $M_{1}$, $F_{1}$, $F_{2}$ and $F_{3}$ calculated in a simulated spherical water site of diameter *D* = 1 nm for protons, helium and carbon ions at different velocities are available in Appendix A.

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
