# Peer review of "Track Structure of Light Ions: The Link to Radiobiology"

_ijms, 2023, doi:10.3390/ijms24065826_

Round 1

Reviewer 1 Report

This article proposes to correlate ionization cluster size distribution quantities to biological damage for light ions. The relevance of this type of study is incontestable, particularly for the optimization of radiotherapy treatments. The paper is clear and well written. I have few comments given the quality of the study presented. However, I think that some points deserve to be detailed or deleted.

- l.59-60 : if I understand correctly your analysis was performed from track structures obtained with an impact parameter set to zero. You tell that the dependence to the impact parameter is “deleted in the notation” but the dependence on ICSD quantities still exist. I think it is therefore important that you discuss the bias introduced by this hypothesis and what to expect from it on the different graphical representations you propose.

- l.335-337: “the total and the differential ionization cross section for light ions was calculated by applying the model of Rudd et al.” This is only for MC-Startrack. As you did Geant4-DNA simulations, one could believe that Geant4-DNA also uses the Rudd model which is true but only below 500 keV. For some energies of the benchmark, the Born model has been used. Be more explicit in this sentence.

- The benchmark with Geant4-DNA as presented is not very meaningful and I don't see what this contributes to your paper which deals with the correlation of ICSD quantities and biological damage. Either remove this aspect or bring it fully into the discussion.

Author Response

Authors thanks Reviewer 1 for the very interesting comments, which lead to an improved version of the manuscript.

Please see the attachment for detailed responses.

Reviewer 2 Report

“ Track structure of light ions: the link to radiobiology” investigates correlation between the distribution of ionizations at the nanometric scale and the probability to induce a radiobiological damage using MC-Startrack Monte Carlo track structure code. This paper is interesting and provides better understanding of radiobiological effects related to the exposure to light ions. However, I have a few comments below: 

1.     Overall English of the paper need to be improved.

2.     Page 1, L43-L46 “Though the description …” need to be rewritten. 

3.     Page 6, L169 “If the target volume is largen..”, “largen” must be changed to “larger”.

4.      Lastly, in M&M, I think it would be a good idea to document the information about the used MC setup according to the recommendations of AAPM TG 268 (https://doi.org/10.1002/mp.12702).

Author Response

We thank reviewer 2 for taking the time to review the manuscript, and for his/her comments, which helped us to write an improved version.

Please see the attachment for detailed answers.

Round 2

Reviewer 1 Report

Thank you for your replies. I look forward to seeing your next paper with a comparison to other codes such as Geant4-DNA.